# An Integrated, Tentative Remote-Sensing Approach Based on NDVI Entropy to Model Canine Distemper Virus in Wildlife and to Prompt Science-Based Management Policies

**DOI:** 10.3390/ani12081049

**Published:** 2022-04-18

**Authors:** Emanuele Carella, Tommaso Orusa, Annalisa Viani, Daniela Meloni, Enrico Borgogno-Mondino, Riccardo Orusa

**Affiliations:** 1Istituto Zooprofilattico Sperimentale Piemonte, Liguria e Valle d’Aosta (IZS PLV) S.C Valle d’Aosta—CeRMAS (National Reference Center for Wildlife Diseases), Località Amerique, 7/C, 11020 Quart, Italy; riccardo.orusa@izsto.it; 2Department of Agricultural, Forest and Food Sciences (DISAFA), GEO4Agri DISAFA Lab, Università degli Studi di Torino, Largo Paolo Braccini 2, 10095 Grugliasco, Italy; tommaso.orusa@unito.it (T.O.); enrico.borgogno@unito.it (E.B.-M.); 3Department of Veterinary Sciences (DSV), Università degli Studi di Torino, Largo Paolo Braccini 2, 10095 Grugliasco, Italy; annalisa.viani@edu.unito.it; 4Istituto Zooprofilattico Sperimentale Piemonte, Liguria e Valle d’Aosta (IZS PLV)—S.C. Ricerca, Piani e Coordinamento Centri di Referenza–S.S. Piani Finalizzati e Coordinamento Centri di Referenza e NRL, Via Bologna 148, 10154 Torino, Italy; daniela.meloni@izsto.it

**Keywords:** CDV, GIS, remote sensing, NDVI entropy, Orfeo Toolbox, Google Earth Engine (GEE), Sentinel-2, red foxes, wolves, badger, beech marten, PCR, Aosta Valley region

## Abstract

**Simple Summary:**

Canine distemper virus (CDV) is a pathogen that affects wildlife with particular regard to Canidae family such as red foxes, wolves, etc. In this study, we focus on CDV outbreaks in the Aosta Valley territory, an alpine region in the NW of Italy which was affected by important waves of this disease during the years 2015–2020 (hereinafter called τ). Ground data are collected on the entire territory at a municipality level. The detection of the canine distemper virus is performed by means of real-time PCR. By adopting satellite remote-sensing data, we notice that CDV trends are strongly related to anomalies in the NDVI entropy changes through (τ). A tentative local model is developed concerning on-the-ground data, helping veterinarians, foresters, and wildlife ecologists enforce management health policies in a One Health perspective.

**Abstract:**

Changes in land use and land cover as well as feedback on the climate deeply affect the landscape worldwide. This phenomenon has also enlarged the human–wildlife interface and amplified the risk of potential new zoonoses. The expansion of the human settlement is supposed to affect the spread and distribution of wildlife diseases such as canine distemper virus (CDV), by shaping the distribution, density, and movements of wildlife. Nevertheless, there is very little evidence in the scientific literature on how remote sensing and GIS tools may help the veterinary sector to better monitor the spread of CDV in wildlife and to enforce ecological studies and new management policies in the near future. Thus, we perform a study in Northwestern Italy (Aosta Valley Autonomous Region), focusing on the relative epidemic waves of CDV that cause a virulent disease infecting different animal species with high host mortality. CDV has been detected in several mammalian from Canidae, Mustelidae, Procyonidae, Ursidae, and Viverridae families. In this study, the prevalence is determined at 60% in red fox (*Vulpes vulpes*, *n* = 296), 14% in wolf (*Canis lupus*, *n* = 157), 47% in badger (*Meles meles*, *n* = 103), and 51% in beech marten (*Martes foina*, *n* = 51). The detection of CDV is performed by means of real-time PCR. All the analyses are done using the TaqMan approach, targeting the chromosomal gene for phosphoprotein, gene P, that is involved in the transcription and replication of the virus. By adopting Earth Observation Data, we notice that CDV trends are strongly related to an altitude gradient and NDVI entropy changes through the years. A tentative model is developed concerning the ground data collected in the Aosta Valley region. According to our preliminary study, entropy computed from remote-sensing data can represent a valuable tool to monitor CDV spread as a proxy data predictor of the intensity of fragmentation of a given landscape and therefore also to monitor CDV. In conclusion, the evaluation from space of the landscape variations regarding the wildlife ecological corridors due to anthropic or natural disturbances may assist veterinarians and wildlife ecologists to enforce management health policies in a One Health perspective by pointing out the time and spatial conditions of interaction between wildlife. Surveillance and disease control actions are supposed to be carried out to strengthen the usage of geospatial analysis tools and techniques. These tools and techniques can deeply assist in better understanding and monitoring diseases affecting wildlife thanks to an integrated management approach.

## 1. Introduction

Canine distemper virus (CDV) belongs to the Morbillivirus genus within the Paramyxoviridae family. It is an enveloped, single stranded, negative-sense RNA virus that encodes for six structural proteins: matrix (M), fusion (F), hemagglutinin (H), nucleocapsid (*n*), polymerase (L), and phosphoprotein (P) [1,2].

CDV is the causative agent of canine distemper (CD), a severe systemic disease, characterized by fever, respiratory and enteric signs, and neurologic disorders [3]. CD is a worldwide distributed viral disease that is highly contagious and has high morbidity and mortality [4]. CDV transmission occurs most commonly through aerosolization of body secretion and excretions as respiratory exudate or urine. Although CDV is quickly inactivated in the environment, the virus can survive at low temperatures and may be transmitted also through direct contact [5]. The canine distemper disease is therefore more prevalent during the cold, humid, rainy season. The colder weather facilities the maintenance and increases the survival time of CDV in the environment [6]. Although the seasonal occurrence of CDV infection is not well defined, there has been observed a significant increase in the number of cases during the winter months and a corresponding decline during the hot seasons [7], when the relative humidity is low and the temperature begins to rise.

CDV has been detected in several mammalian species in the families Canidae, Mustelidae, Procyonidae, Ursidae, and Viverridae. The infection has also been reported in captive and free-ranging large felids, in captive Japanese primates, and in Siberian seals [3]. CD is considered an important threat to the conservation of different wild carnivore species and has contributed to the population decline of several wild animals [8]. However, the domestic dog (*Canis lupus* familiaris) is still considered to be the main reservoir of CDV whose spread among domestic and wild animals is affected by the dynamics between their populations, genetic characteristics of the virus, host receptors, and other factors that are not fully understood [8]. Nevertheless, the virus may circulate in wildlife thanks to a complex system where different interconnected populations may function as a single reservoir, even when the CDV circulation in domestic dogs is low, in the same study area [1,4]. These wild species that favor the spread of the virus in the wild ecosystems are named “meta-reservoir” [4,9,10].

Epidemiological data from Europe about CDV seroprevalence in wildlife, particularly in red foxes (*Vulpes vulpes*), revealed different rates in wild animals, with a prevalence ranging from 4% to 30% in countries such as Spain, Portugal, Italy, and Germany [4]. Although the canine distemper virus is mainly spread by infected domestic dogs, that in turn trigger the onset in wild animals, the CDV transmission from wildlife to domestic canids is also possible [4]. The Alpine region of Northeastern Italy was hit by a canine distemper outbreak affecting the wild carnivore population in 2006. Several red foxes, badgers, and stone martens tested positive for CDV infection. The canine distemper virus then reached the pre-Alps and urbanized areas in 2007, thus posing a risk to pets [8].

A significant difference in numbers of seropositive foxes between urban, suburban, and rural areas has been observed, suggesting that urban sprawl plays a role in CDV transmission. It has been assumed that human population density is linked to the density of domestic dogs. This in turn may affect the spread of CDV in wild carnivores, that share the same habitats as domestic animals. Therefore, an association between urban sprawl and CDV transmission in wild animals should be considered [11].

The remote sensing could support comprehension of the role of environmental patterns in conditioning the CDV patho-system, such as for the infectious keratoconjunctivitis (IKC) previously investigated in another study [12].

The landscape is an ecological term that refers to a complex system generated by human and natural interactions. The spatial patterns of various land use in a landscape show historical and present processes that shape the landscape’s dynamics and organization as well as the capacity of disease spreading [13]. The understanding of these dynamics has also become crucial in the face of the effects of climate change [14,15,16].

Nowadays, remote sensing missions realized by several public and private agencies allow for the monitoring at different space-temporal resolution the changes that occur to the environment and the identification of their magnitude [17,18,19,20]. Several studies in the public health sectors used GIS to describe environmental conditions, especially concerning vector-borne disease in Africa and Asia [21,22,23,24,25,26,27,28]. Nevertheless, the ongoing research focused on the environment and the application of free satellite missions, such as those of the Copernicus program, are few in the veterinary research [12]. Human and veterinarian medicines are still far to reach consistent studies based on a One Health approach, even if the relationship between the environment and diseases is well known [29,30,31]. SARS-CoV, like many other zoonoses, has demonstrated that both animal and human health are deeply linked to ecosystem health. Nowadays, the relationship between environmental patterns and disease spread should be investigated to assess the best way to prevent the spread of diseases in ecosystems. Following the path opened in a previous study [12], we focus on the potential of remote sensing to achieve a better understanding of the relationships between the environment and CDV at the landscape level.

The considered study area is the Aosta Valley region located in the Northwestern Italian Alps. A new analysis approach is adopted to assess the relationship between land use and CDV spread. Land use changes and urban sprawl are discussed as possible predisposing factors for the spread of canine distemper, among the Alpine wild carnivores [32,33]. To better understand these preliminary results, we introduce a possible novelty in wildlife CDV knowledge, developing a tentative local forecasting model of CDV spread based on ecological open-source remote-sensing data. We use a spatiotemporal analysis of entropy-related indices of Normalized Difference Vegetation Index (NDVI) time series, focusing on its anomalies to investigate the interaction of order and disorder in ecological landscapes (hereinafter called EL). This index could be helpful with complex systems analysis and the assessment of EL changes. In fact, EL variations seem to have a huge role in the condition of CDV spread. Our overarching goal is to demonstrate a possible relationship between CDV spread and entropy change anomalies. Therefore, our main aim is to depict the presence of CDV in Aosta Valley as well as explore a possible relationship between entropy variations and CDV spread, providing a useful tool for veterinarians to enforce ecological studies and new management policies.

## 2. Study Area

The Aosta Valley Autonomous Region, located in the Northwestern Italian Alps, was considered in this study (see Figure 1). The National Reference Center of Wildlife Disease (hereinafter called CeRMAS) has its headquarters in this Italian–Francophone territory. It is worth remembering that CeRMAS has a large dataset of zoonoses and wildlife diseases that has been collected here for several years or longer, and this exists at the Italian level too.

Despite the small size of this Alpine region, the presence of wildlife, and biodiversity in general, is very high in general, making it an open-air laboratory for many scientific areas [34]. 

## 3. Materials and Methods

Sample collection. The brain, the lung, the bladder, and the spleen of different wild animals were collected between 2015 and 2020 in Aosta Valley, an Italian region located in the northwestern Alps between Switzerland and France. The specimens were obtained from different wild species found dead or shot during the hunting seasons, previously collected thanks to the Corpo Forestale della Valle d’Aosta (Forest Guards). Subsequently the samples were placed in sterile tubes and stored at −80 °C before being tested. The wild animals investigated for the presence of CDV, through real-time polymerase chain reaction (RT-PCR) were red fox (*Vulpes vulpes*, *n* = 281), wolf (*Canis lupus*, *n* = 18), beech marten (*Martes foina*, *n* = 47), and badger (*Meles meles*, *n* = 101).

Deoxyribonucleic acid (DNA) extraction and real-time PCR conditions. The brain, the lung, the bladder, and the spleen of each wild animal included in the study were individually weighed (50 mg) and placed in Precellys CK 28 tubes containing zirconium oxide beads (VWR, Radnor, PA, USA). Subsequently, 1 mL of QIAzol Lysis reagent (Qiagen, Hillden, Germany) was added to the tubes containing the samples, which in turn were homogenized by means of the tissue homogenizer Precellys 24 (Bertin instrument, Montigny-le-Bretonneux, France). The RNeasy Lipid Tissue Mini kit (Qiagen, Germany) was then used for the extraction of total RNA, following the manufacturer’s instructions. The detection of canine distemper virus was performed by means of real-time PCR. All the analyses were carried out by means of the StepOnePlus™ Real-Time PCR System (Thermofisher, Waltham, MA, USA). The CDV detection employed the TaqMan approach, targeting the chromosomally gene for phosphoprotein (gene P) that is involved in the transcription and replication of the virus. The real-time PCR protocol was applied, according to Scagliarini et al. [35], using the SuperScript III Platinum One-step Quantitative RT-PCR System (Invitrogen, Carlsbad, CA, USA) and the primers and probe manufactured by Thermofisher. The PCR reaction mixture consisted of 500 nM of the forward (5′-AACCACAGGCATGCAGGAA-3′) and the reverse (5′-GCCGACATAGTTTGATCCTTTT-3′) primer, 200 nM of the probe (5′-FAM-CTCTCAGAATCTCGATGAATCACACGAGCC-3′-TAMRA), and 5 μL of template, in a total volume of 25 μL. The PCR cycling conditions consisted of an initial reverse transcription step with a temperature of 50 °C for 15 min, followed by 45 cycles at 95 °C for 15 s and 60 °C for 30 s. An internal positive control (IPC) was added to the PCR reaction mixture, using Armored RNA (Ambion, Austen, TX, USA), a pseudoviral particle containing the NS5-2 region of the West Nile virus genome packaged inside bacteriophage coat proteins [36]. The primers and the probe for the NS5-2 region were manufactured by Thermofisher. If the IPC cycle threshold value was higher than 39, the template was then diluted at 1:10 with ddH_2_O and subjected to a second amplification. Prevalence data were finally computed by Equation (1) with reference to the entire regional territory for all the monitored years (2015–2020).
(1)$$\mathrm{Pr}\;=\frac{C}{P}\times 100$$
where Pr = disease of the CDV in wild animals (%), C = number of wild animals who tested positive for CDV, and P = number of wild animals examined.

Earth Observation Data processing and GIS. The EO Data were obtained and processed in Google Earth Engine (hereinafter called GEE) [37] and the Sentinel-2A data were used. European Space Agency (ESA) Copernicus Sentinel-2 (hereinafter called S2) is a wide-swath, high-resolution, multi-spectral imaging satellite mission supporting Copernicus Land Monitoring studies, including the monitoring of vegetation, soil, and water cover, as well as observation of inland waterways and coastal areas. The spatial resolution ranges between 10 and 60 m depending on the band considered. The temporal resolution of 10 days (S2A and S2B, respectively; combined they have 5 days of temporal resolution) and a spectral resolution as described in Table 1.

The S2 L2 data were downloaded from ESA Copernicus Scihub (https://scihub.copernicus.eu/ last access 5 March 2022). They were computed by running sen2cor (a script provided by European Space Agency (ESA) working in many platforms such as ESA SNAP tool v.7.0.0, that allows for the performance of ortho-rectification and applies the radiative transfer model to calibrate the imagery to surface reflectance—SR).

For this study, the COPERNICUS/S2 collection in GEE was adopted because it provides the temporal coverage requested for performing the whole necessary analysis ranging from 2015 to 2020. It is worth remembering that GEE provides calibrated SR S2 data only from 28 March 2017 to present. COPERNICUS/S2 asset contains 12 UINT16 spectral bands representing top of the atmosphere (TOA) reflectance scaled by 10,000. In addition, three QA bands are present where one (QA60) is a bitmask band with cloud mask information. Clouds can be removed by using COPERNICUS/S2_CLOUD_PROBABILITY.

The adopted collection has pre-processed Level-1C products that include radiometric and geometric corrections including orthorectification and spatial registration on a global reference system with sub-pixel accuracy. In this study, clouds were masked by applying QA60 to the entire image collection while also considering cirrus. To do this, on the entire collection, the Fito Principe library, available freely on GEE, was adopted. This extensive package requires “users/fitoprincipe/geetools:cloud_masks” which permits iteratively masking out all the clouds in S2 images (https://github.com/fitoprincipe/geetools-code-editor) (accessed on 14 March 2022).

To get BOA (Bottom of the Atmosphere) data, the whole TOA collection was calibrated in GEE adopting the Sensor Invariant Atmospheric Correction (SIAC) algorithm realized by the Department of Geography of the University College of London [38,39,40].

This atmospheric correction method uses MODIS MCD43 BRDF product to get a coarse resolution simulation of Earth’s surface. A model based on MODIS PSF is built to deal with the scale differences between MODIS and Sentinel-2 as well as Landsat 8 satellite missions. In addition, the ECMWF CAMS prediction was considered in the method, as a prior for the atmospheric states, coupling with 6S model to solve for the atmospheric parameters. The algorithm in its present form does not include topographic correction considering the ground as a homogeneous surface. Since Aosta Valley has a geomorphological complex territory in a prevalently mountainous region, a topographic correction was performed by adding in the SNIC algorithm to the DTM to consider the BRDF (Bidirectional Reflectance Distribution Function) effects. The Aosta Valley DTM provided by the regional cartographic resources (https://geoportale.regione.vda.it/ accessed 5 March 2022) was obtained during a lidar flight that occurred in 2008 and it has a native ground sample distance of 2 m. Thus, it was resampled to 10 m in SAGA GIS v.8.0.0 [41] as a reference grid target of the S2 stack of images adopted.

The altitude of each sample was obtained directly by the data thanks to the fact that it was determined during the discovery of the animal. In the case of the samples without altimetric reference, the altitude was obtained starting from GIS analyses (intersect with DTM) which provided for the assignment of a centroid on the geometry of the municipality or municipal fraction indicated by the forestry officials at the time of the discovery. The analysis was performed in QGIS [42].

Entropy computation. Starting from the S2 filtered and calibrated stacked collection ranging from 2015 to 2020 for each year, the median NDVI was computed as follows at a pixel level [43]:(2)$$\mathrm{NDVI}=\frac{\rho_{\mathrm{NIR}}-\rho_{\mathrm{RED}}}{\rho_{\mathrm{NIR}}+\rho_{\mathrm{RED}}}$$
where ρ_NIR_ and ρ_RED_ are S2 B8 and B4 reflectance values, respectively.

Yearly NDVIs had thresholds in SAGA GIS v.8.0.0 to define two types of land cover. A vegetation cover (including agriculture, forests, grasslands) and an urban-anthropic cover (including built-up areas and bare soils and low vegetation). Water bodies and water courses as well as glaciers and rocks were masked out by adopting local cartography (the official Land Cover of Aosta Valley that is also going to be published in a scientific journal).
(3)NDVIt=NDVI>0.3 

The masking adopted is reported in the image below (please see Figure 2).

Starting from this spectral index, we computed the H_NDVIt_ as follows:(4)$$H_{\mathrm{NDVIt}}=-\overset{N-1}{\underset{i=0}{\sum }}\overset{N-1}{\underset{j=0}{\sum }}{\mathrm{NDVIt}}_{i,j}\log\left({\mathrm{NDVIt}}_{i,j}\right)$$
where NDVIt_i,j_ is the NDVI value at the i-th row and j-th column in the local square window measuring N pixels. For this study, a kernel window size of 10 **×** 10 pixels was adopted.

The entropy was computed on Orfeo Toolbox vers. 8.0.0, an open-source tool for remote-sensing analysis. In particular, the Haralick texture features function was adopted [44,45]. This application computes Haralick, advanced, and higher order texture features on every pixel in the selected channel of the input image, in this case the NDVI.

NDVI entropy (H_NDVI_) was thought to be a suitable image texture parameter capable of distinguishing the intensity of changes in both the natural (woods, grasslands, agriculture) and anthropic systems. As demonstrated by [46], entropy is also a valuable tool to define agricultural production systems. Starting from these premises, we considered that abrupt changes (higher than 0.05 H_NDVIt_ to exclude internal vegetation variation due to biological factor) cause an effect on the landscape and therefore on wildlife movements and the possibility of their interaction. Lower values of H_NDVIt_ means order in the landscape while higher values define a great degree of disorder and therefore entropy in the system. According to [32], changes in land use as well as land cover (hereinafter called LULC) involve a major risk of CDV spread. Starting from this study’s preliminary results, we considered H_NDVIt_ space and temporal distribution as a possible factor of disease spreading. In the Aosta Valley landscape, changes in LULC evaluating NDVI entropy were more varied and dispersed, resulting in increased H_NDVIt_ of these patches. With these conditions, the most important component of the workflow was detecting the anomaly of entropy on a reference time to compare the mean H_NDVIt_ of the territory (assumed as reference typical conditions) to differences that occurred on a given year. The anomaly of entropy (AH) was computed as follows:(5)$$\mathrm{AH}=\frac{H_{\mathrm{NDVIt}}}{{\mu H}_{\mathrm{NDVIt}}}$$
where H_NDVIt_ is the entropy of NDVI in a given time (in the example, a given year) and μH_NDVIt_ is the mean NDVI entropy computed in the reference time 2015–2020. Statistics such as the AH were computed in SAGA GIS vers. 8.0.0, RStudio [47,48,49], and PAST [50,51] and the map visualization was conducted on QGIS v.3.16.4. In this analysis, the DTM was used to assess a possible relationship between the CDV and the altitude starting from ground data.

## 4. Results

Canine distemper virus prevalence (Pr) was computed according to the available ground dataset, by Equation (1) after detecting the presence of the CDV in the wild animals, collected by Corpo Forestale della Valle d’Aosta, by means of real-time PCR. The numbers of collected samples can be considered statistically significant by applying statistical inference, taking into account the population data (available https://www.regione.vda.it/corpoforestale/competenze/fauna_selvatica_i.aspx, last access on 9 March 2022) to all species considered, apart from the wolf. Values reported in Table 2 refer to the period 2015–2020. Despite the fact that the number of samples is not uniform to all wild species considered in this study, the red foxes that were positive for CDV weighed on the entire analyzed population, resulting in the highest occurrence followed by mustelids. To assess CDV trends through time, the yearly prevalence was computed at a regional level for the entire period starting from 2015 to 2020 (see Table 3).

To perform analysis taking into account the NDVI entropy, CDV prevalence was grouped into a single one value without considering the single species, starting from the following assumptions: (1) changes in LULC affect movements and possible interaction in the whole wildlife population, (2) the CDV spread involves all canine families, (3) some species, such as wolves, do not have a statistically significant number of samples to perform statistic inference, and (4) GIS and remote-sensing analysis usually works better with a high number of ground data especially in developing a tentative local model.

Therefore, the CDV data were grouped together without considering the species for each year as follows:

A tentative local model simply based on the CDV outbreak throughout the year was developed (see Figure 3). Even if the determination coefficient is high, adopting a 3rd degree polynomial order equation, a non-parametric trend test (Mann–Kendall trend test) was performed to assess the *p*-value. The trend observed is not significant because the *p*-value is higher than 0.05. We know that this model is not robust enough and that it is based on a short time series due to the lack of past data. Nevertheless, it represents a preliminary tentative tool to better understand how the disease affects wildlife in a specific area, and we hope that it can help local veterinarians, as well as researchers, to improve studies at a different level.

Before performing entropy analysis, we analyzed if altitude was able to condition the CDV spread. We started from the assumption that in Aosta Valley the most intense changes in LULC in terms of surface occurred in the bottom of the valley. Therefore, we performed a regression considering the positive CDV wildlife with respect to the altitude considering all the analyzed years. We know that wildlife moves but it is true that wildlife hunting activities and their feedings generally happen in transitional ecological patches, such as anthropic and natural patches (this is particularly true for red foxes). Therefore, changes in these areas due to agriculture, urbanization, forestry, and natural disturbances deeply affected wildlife interaction and consequently CDV spread. Below we report the tentative model realized considering the altimetry and in particular the CDV positive samples in the period 2015–2020 (see Figure 4).

The trend to model the relationship is exponential and the major positive animals are located at lower altitude and appear to decrease when the altitude increases. This is likely due to a lower possibility of interaction between animals and a context of sudden variation of the landscape due to anthropogenic activities. Therefore, these results seem to confirm the analyses carried out considering entropy.
(6)$$\mathrm{CDVpr}\;=44.76{\;e}^{-0.002q}$$
where CDVpr is the CDV prevalence while q the quote (m) detected during the collection of the wildlife sample.

Statistical analysis has shown a coefficient of determination R^2^ = 0.75 and a statistically significant trend *p*-value < 0.005. In Aosta Valley, this model that considers the quote is strong enough to be adopted by veterinarians, ecologists, and foresters to model local CDV spread. This analysis allowed us to assess the previous assumption and focus the research on the variation in terms of distribution of the patches, detecting the variations in terms of increased disorder in the components of the landscape under study. By computing from EO data the mean entropy in the period ranging from 2015 to 2020 and assessing for each year the anomaly including both the vegetational and urban component, a tentative forecasting general linear model GLM was developed as follows (see Figure 5). In order to better explain the entropy analysis, we computed the anomalies through the year. In particular, for each year, we computed the entropy and then we calculated the mean entropy for the whole period ranging from 2015 to 2020 as reference.

The tentative local model realized has shown that there is a strong correlation between changes in landscape patches and the CDV prevalence. Statistical analysis has shown a coefficient of determination R^2^ = 0.85. Nevertheless, a significant non-parametric test (Mann–Kendall) applied to data on PAST software has shown a *p*-value > 0.05. Therefore, to evaluate the quality of these preliminary results, only cross-validation with the future ground date with this empirical model may help to better understand the solidity of the realized model. Below we report the tentative remote-sensing approach based on NDVI entropy to try to model canine distemper virus (CDV) in the Aosta Valley territory.
(7)CDVpr=540.75∗AH −487.76 
where CDVpr is the CDV prevalence in the tentative local model performed in Aosta Valley region (%) and AH is the anomaly of entropy computed according to Equation (5). In conclusion, below we report a flowchart of the procedure adopted and the relative maps of H_NDVIt_ obtained (see Figure 6).

## 5. Discussion

CD is a multisystemic disease and its clinical severity both depends on the strain pathogenicity and the host immune status [1]. Although host range expansion has been reported, the domestic dog is still regarded to be the principal viral reservoir and the risk factors associated with CDV exposure in wildlife are not fully understood [16]. The canine distemper virus appeared to circulate throughout Northwestern Italy between 2013 and 2015, with the highest prevalence in the Aosta Valley region. It has been suggested that environmental factors such as the geographical characteristics of Northwestern Italy could be a key factor in the spread of CDV [1]. In addition, a high lethality rate of CD has both been reported in mustelids and in red foxes. Although the spatial distribution of the red fox population has not been reported in the northwest of Italy, this wild animal has the widest geographic distribution among the canids. Therefore, a potential role of the red fox in the spread and maintenance of CD in this study area is possible [1]. Based on the results of this study, the circulation of the canine distemper virus in the wildlife located in Aosta Valley did not end in 2015 but continued until at least 2020. The prevalence of CDV in wild carnivores appears to be consistent with that reported by Di Blasio et al. [1] and no significant difference in susceptibility to canine distemper virus was observed among the mustelids investigated (beech marten 51% and badger 47.5%), except for the canids (red fox 58% and wolf 37.5%).

This CDV latency that was shown to have lasted until 2020 may be attributable to changes in homogeneity and inhomogeneity across the landscape. Changes in LULC due to anthropogenic activities and natural disturbances, especially in urban–natural interface contexts, have strong repercussions on the spread of the pathogen. This is due to the fact that the variation of ecological corridors can affect the possibility of animal interaction. Similarly, the risk of interaction between domestic and wild animals is greater in urbanized contexts, which in the Aosta Valley territory are mainly concentrated in the bottom of the valleys. This factor seems to explain the close relationship between altitude and CDV positivity. The altitude gradient seems to be a conditioning factor for wildlife positivity. This aspect seems to corroborate the thesis proposed in the experimental study of the following work in which the anomaly in the entropy of NDVI appears to be a good predictor of the risk of CDV spread. In fact, the areas with higher entropy (disorder and inhomogeneity) are in correspondence to anthropized areas and the urban–natural interface. Think of the different agronomic cultures, types, and distribution of urbanized areas.

As shown in previous studies on the same topic [13,52,53], it is difficult to find the best spatial resolution for studying patterns at the ecosystem level, yet it is critical to comprehend and interpret emergent patterns because no single spatial resolution is appropriate for all ecological phenomena [10,12].

The results obtained confirmed the findings performed by [32,33,54,55] and hopefully they open new perspectives in future studies concerning CDV environmental dynamics. It would certainly be interesting to increase and deepen the study in other areas about the role of altimetry and entropy by adopting a remote-sensing approach.

Finally, the evaluation from space of the landscape variations with particular regard to wildlife ecological corridors due to anthropic or natural disturbances may help veterinarians and wildlife ecologists to enforce management health policies in a One Health perspective by pointing out time and spatial conditions of interaction between wildlife. Surveillance and disease control actions have to be carried out to strengthen the usage of geospatial analysis tools and techniques that can strongly better the understanding and monitoring of diseases affecting wildlife thanks to an integrated management approach favored by technology transfer in the various sectors of knowledge.

## 6. Conclusions

CDV spread proved to be conditioned by changes along an altimetry gradient and LULC changes with a focus on the distribution on the landscape in terms of order and disorder. In particular, the evaluation of anomalies in the Normalized Difference Vegetation Index (NDVI) entropy on the vegetational and urban component seemed to be a good predictor. Today, the huge amount and availability of free and global remote-sensing data is certainly a valid “systemic” tool for risk analysis and modelling that can support ordinary diagnostic techniques, allowing continuous monitoring of the effects of LULC changes in mountain and wilderness environments. Therefore, GIS processing considering quote and time series analysis in order and disorder in EL can be a good way to monitor CDV in wildlife. As a preliminary result, we found a possible relationship between CDV spread and entropy change anomalies at a local level. We know that further studies must be performed to establish the quality of the present study and obtain evidence in scientific literature. Nevertheless, these tentative models obtained may represent another step to perform a One Health approach. In fact, we hope that GIS instruments and procedures coupled to ordinary health techniques may help veterinarians, foresters, and urban planners to enforce ecological studies and new management policies. In conclusion, surveillance and disease control actions should be carried out by strengthening the usage of geo-spatial analysis tools and techniques. This can help significantly in the monitoring of diseases affecting wildlife thanks to an integrated management approach favored by technology transfer in the various sectors of knowledge, including veterinary medicine.

## Figures and Tables

**Figure 1 animals-12-01049-f001:**
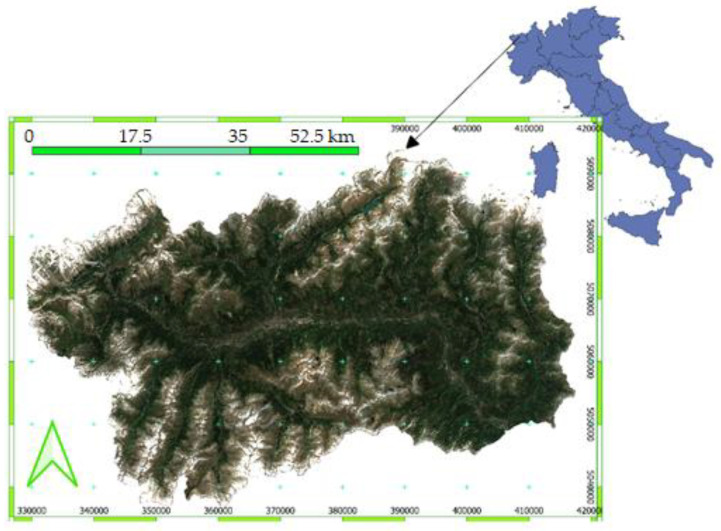
Study area. The Aosta Valley region in NW Italy. Reference system ED50-UTM 32 N.

**Figure 2 animals-12-01049-f002:**
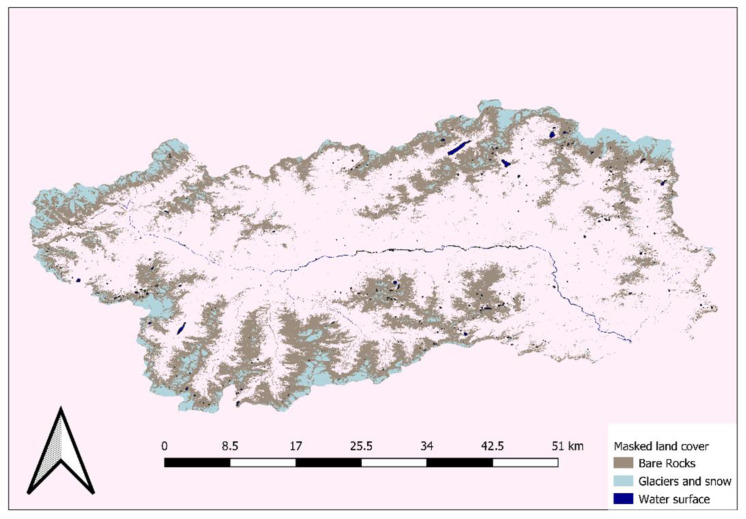
Surfaces not included in the computation of NDVIt Entropy and that are therefore masked. Reference system WGS84.

**Figure 3 animals-12-01049-f003:**
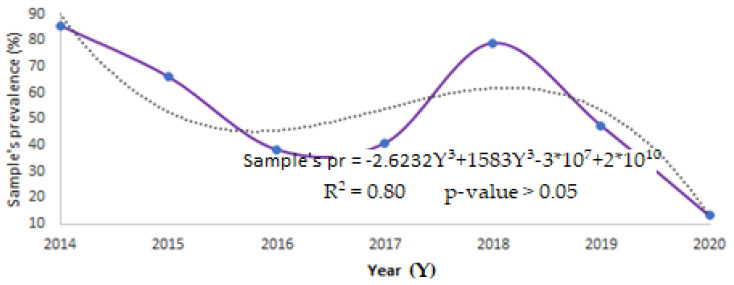
CDV trends in Aosta Valley.

**Figure 4 animals-12-01049-f004:**
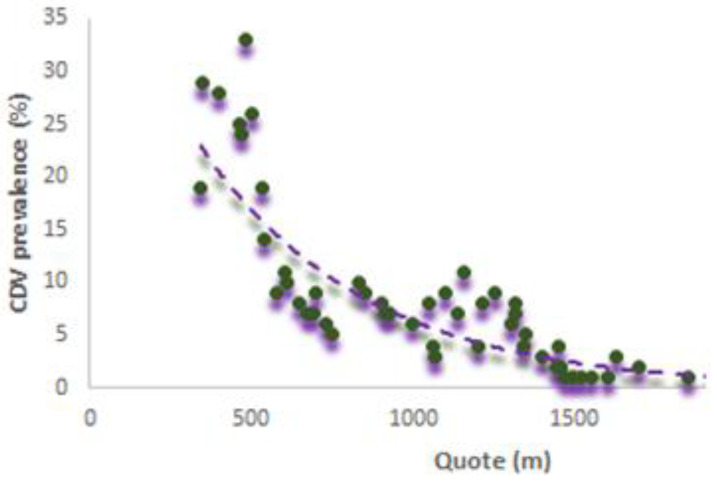
CDV trends in Aosta Valley.

**Figure 5 animals-12-01049-f005:**
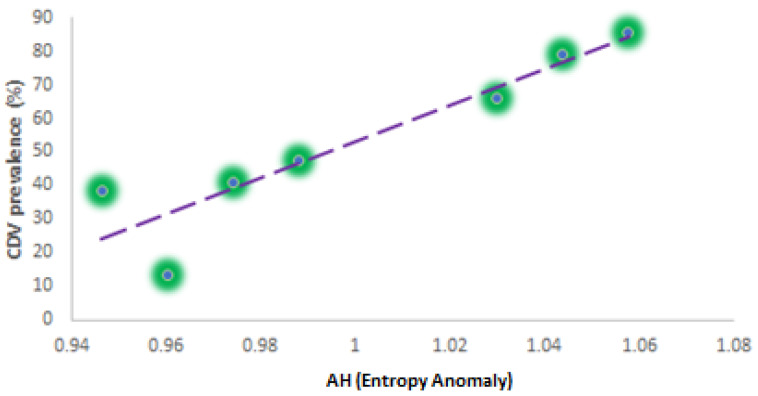
GLM between anomalies in NDVI entropy and CDV spread (data were grouped annually considering the entire Aosta Valley territory).

**Figure 6 animals-12-01049-f006:**
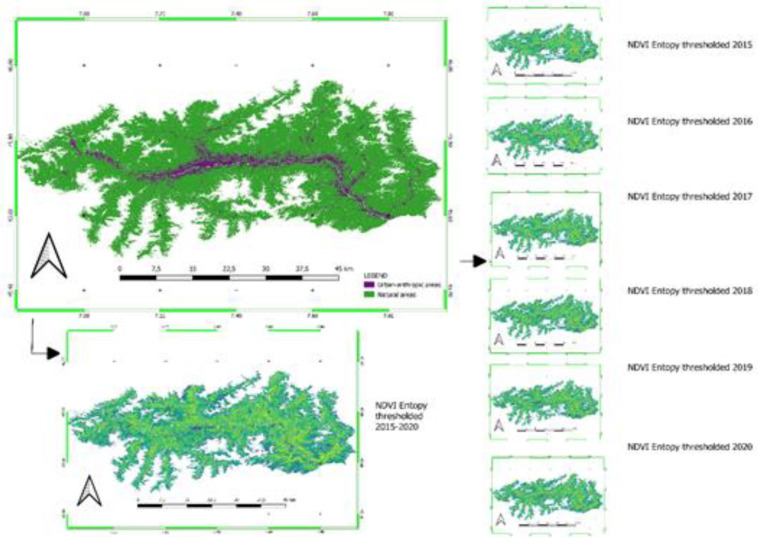
H_NDVIt_ maps adopted and calculated at a pixel level, grouped into two classes, and finally merged into a final one. Reference system WGS84.

**Table 1 animals-12-01049-t001:** Sentinel-2 bands, ground sample distance, and wavelengths.

Sentinel-2 Bands (B*)	Central Wavelength (nm)	Bandwidth (nm)	Geometric Resolution (m)
B1–Coastal aerosol	442.7	21	60
B2–Blue	492.4	66	10
B3–Green	559.8	36	10
B4–Red	664.6	31	10
B5–Vegetation red edge	704.1	15	20
B6–Vegetation red edge	740.5	15	20
B7–Vegetation red edge	782.8	20	20
B8–NIR	832.8	106	10
B8A–Narrow NIR	864.7	21	20
B9–Water vapor	945.1	20	60
B10–SWIR–Cirrus	1373.5	31	60
B11–SWIR	1613.7	91	20
B12–SWIR	2202.4	175	20

**Table 2 animals-12-01049-t002:** CDV prevalence in the Aosta Valley region. Data refers to the years 2015–2020.

Animal Species	CDV Prevalence (%)	Number of Samples Analyzed	Positive for CDV
red fox	58	281	164
wolf	37.5	18	3
beech marten	51	47	24
badger	47.5	101	48

**Table 3 animals-12-01049-t003:** CDV prevalence in the Aosta Valley region. Data refers to each year from 2015 to 2020.

Year	CDV Prevalence (%)
2014	85.7
2015	66.2
2016	38.5
2017	40.8
2018	79.0
2019	47.5
2020	13.2

## Data Availability

The CDV data can be requested from CeRMAS. The geospatial data can be freely downloaded from Google Earth Engine at this link: https://code.earthengine.google.com (accessed on 14 March 2022) or from the official ESA (European Space Agency) Scihub, reachable at this link: https://scihub.copernicus.eu/ (accessed on 14 March 2022).

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
