# Peer review of "An Integrated, Tentative Remote-Sensing Approach Based on NDVI Entropy to Model Canine Distemper Virus in Wildlife and to Prompt Science-Based Management Policies"

_animals, 2022, doi:10.3390/ani12081049_

Round 1

Reviewer 1 Report

Dear Authors,

Information in this manuscript is definitely beneficial for researchers whom it concerns, veterinarians and wildlife ecologists. It is well written, however, I have some remarks. I suggest that this manuscript needs a minor revision before the publication. Please, see the comments and concerns in the attached pdf file.

Reviewer 2 Report

General comments:

I think the work presented in the paper is relevant for the field and can be beneficial to the control of diseases in the wildlife animal population. Their results showed they were able to develop a model to predict the presence of canine distemper in the wildlife population in the Aosta Valley Region. This model will help to develop and enforce health policies to prevent and control the spread of canine distemper.

The paper is well-written; however, the authors use some colloquialism, especially in the simple summary and abstract sections, it is recommended to language expert revision of the paper.

Specific comments:

Line 26:” to anomalies in the NDVI Entropy changes through (τ).” The sentence sounds incomplete. Does the author mean through time or a through the years 2015-2020?

Line 42-43: “All analyses were carried out thanks to the TaqMan approach” This sentence should be re-written: All analyses were done using TaqMan approach,

Line 68-70: “CDV transmission probably occurs most commonly through aerosolization of a respiratory exudate containing the virus, besides other body excretions and secretions, including urine, that can result in infection if aerosolized.” It is a long sentence that should be re-written: CDV transmission occurs most commonly through aerosolization of body secretion and excretions as respiratory exudate or urine.

Line 189-190: The author should add the primer sequences used on the material and methods section.

Line 190-192: I did not understand why the author would dilute 1:10 the cDNA sample of Ct values higher than 39. Usually, high Ct values are associated with samples that have a low concentration of the target gene mRNA/cDNA. Can the author explain the rationale behind this procedure?

Line 195-196: the equation of disease prevalence is correct. However, the population examined in this paper was only animals found dead. This percentage of animals with the disease of the examined animals may not represent the whole population. Because canine distemper usually causes high mortality in the population, the high prevalence of dead animals positive for the disease will be super estimated. This should be addressed in the result or discussion part.

Figure 5 legend: It should add the name of the x-axis (anomaly of entropy computed) to the legend.
